# PROMPT-CONSISTENCY IMAGE GENERATION (PCIG): A UNIFIED FRAMEWORK INTEGRATING LLMS, KNOWLEDGE GRAPHS, AND CONTROLLABLE DIFFUSION MODELS

## ABSTRACT

The rapid advancement of Text-to-Image(T2I) generative models has enabled the synthesis of high-quality images guided by textual descriptions. Despite this significant progress, these models are often susceptible in generating contents that contradict the input text, which poses a challenge to their reliability and practical deployment. To address this problem, we introduce a novel diffusion-based framework to significantly enhance the alignment of generated images with their corresponding descriptions, addressing the inconsistency between visual output and textual input. Our framework is built upon a comprehensive analysis of inconsistency phenomena, categorizing them based on their manifestation in the image. Leveraging a state-of-the-art large language module, we first extract objects and construct a knowledge graph to predict the locations of these objects in potentially generated images. We then integrate a state-of-the-art controllable image generation model with a visual text generation module to generate an image that is consistent with the original prompt, guided by the predicted object locations. Through extensive experiments on an advanced multimodal hallucination benchmark, we demonstrate the efficacy of our approach in accurately generating the images without the inconsistency with the original prompt.

## 1 INTRODUCTION

The rapid advancement of Text-to-Image (T2I) generative models has revolutionized the field of computer vision, enabling the synthesis of high-quality images guided by textual descriptions. These models, such as DALL-E DAL (2023); Ramesh et al. (2021), Stable Diffusion Podell et al. (2023), and GLIDE Nichol et al. (2021), have shown remarkable progress in generating visually appealing and semantically relevant images. However, despite their impressive performance, these models often generate contents that contradict the input text, posing significant challenges to their reliability and practical deployment.

Inconsistencies between the visual output and textual input can manifest in various forms, such as mismatched object attributes (First image in Figure 1), inaccurate object placement or count (Fourth image in Figure 1), illegible or incorrect text within the image (Second image in Figure 1), and the inability to accurately depict real-world entities (Third image in Figure 1). These inconsistencies, also known as hallucinations, can severely impact the usefulness and trustworthiness of the generated images, especially in domains where accuracy is crucial, such as medical imaging Kazerouni et al. (2023); Liu et al. (2024b), autonomous vehicles Liu et al. (2024a), and criminal investigation Nowroozi et al. (2021).

Existing methods have attempted to address these challenges by improving the alignment between the input text and the generated image. Attention-based approaches Ho et al. (2020); Song et al. (2020) have been proposed to better capture the relationships between words and visual features, while adversarial training techniques Frolov et al. (2021) have been employed to enhance the realism and consistency of the generated images. However, these methods often struggle with complex scenes and fail to address the specific types of inconsistencies mentioned earlier.

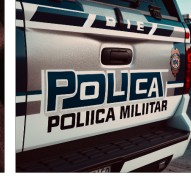 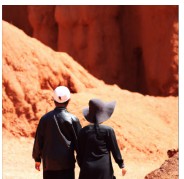 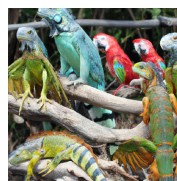

a **brown** jacket and a **black** hat.    A police car reads **"POLICIA MILITAR"** across the back.    Give me a picture about "夫妻肺片"(a dish commonly known as "Spicy Sliced Beef and Ox Tripe" in English).    **Eight** colorful parrots perched on branches near a green iguana.

Figure 1: Selected samples generated by DALL-E 3. Each image represents one specific hallucination type. The inconsistency part for each image is highlighted in red.

To tackle these limitations, we introduce Prompt-Consistency Image Generation (PCIG), a novel diffusion-based framework that significantly enhances the alignment of generated images with their corresponding descriptions. PCIG addresses three key aspects of consistency: (1) general objects (GO), ensuring accurate depiction of object attributes and placement; (2) text within the image (TEXT), generating legible and correct text; and (3) objects that refer to proper nouns existing in the real world (PN), which cannot be directly generated by the model.

Our framework leverages state-of-the-art techniques in natural language processing and computer vision. We first employ large language models (LLMs) OpenAI (2023) to extract objects from the input prompt and construct a knowledge graph to predict the locations of these objects in the generated image. LLMs, such as GPT-3 Brown et al. (2020) and BERT Devlin et al. (2018), have shown remarkable capabilities in understanding and generating human language. By integrating LLMs into our framework, we enable a deeper understanding of the prompt and its relationships, guiding the subsequent image generation process.

Next, we utilize a controllable diffusion model Li et al. (2023); Wang et al. (2024a); Zhou et al. (2024) to generate an image consistent with the original prompt, guided by the predicted object locations. Controllable diffusion models allow for more fine-grained control over the image generation process by incorporating additional constraints or conditions. For general objects (GO), the model focuses on accurate attribute depiction and spatial arrangement. To handle text within the image (TEXT), we incorporate a visual text generation module Tuo et al. (2023); Ma et al. (2023); Yang et al. (2024) that specializes in rendering legible and semantically correct text. Recent advances in visual text generation have shown promising results in producing realistic and readable text in images. Finally, for objects referring to proper nouns (PN), we propose a novel approach that searches for representative images of the entities and seamlessly integrates them into the generated image.

Through extensive experiments on an advanced multimodal hallucination benchmark Chen et al. (2024b), we demonstrate the efficacy of PCIG in generating images that align with the original prompt, significantly reducing inconsistencies across all three key aspects. Our unified framework achieves state-of-the-art performance, outperforming existing T2I models in terms of object hallucination accuracy, textual hallucination accuracy, and factual hallucination accuracy.

**Contributions.** (1) We introduce PCIG, a novel framework that integrates LLMs, knowledge graphs, and controllable diffusion models to generate prompt-consistent images. (2) We propose a comprehensive approach to address three key aspects of consistency: general objects, text within the image, and objects referring to proper nouns. (3) We conduct extensive experiments on a multimodal hallucination benchmark, demonstrating the superiority of PCIG over existing T2I models in terms of consistency and accuracy. (4) We provide insights into the effectiveness of integrating LLMs and knowledge graphs for prompt understanding and object localization in the image generation process.

## 2 RELATED WORK

This section presents an overview of Text-to-image Diffusion Models, Controllable Image Generation, LLM-assisted Image Generation, and Knowledge Graph and LLM. The background and related work about the Visual Text Generation and is provided in the Appendix A.1.

## 2.1 Text-to-image Diffusion Models

Denoising Diffusion Probabilistic Model Ho et al. (2020); Song et al. (2020) and its subsequent studies Ho & Salimans (2022); Ramesh et al. (2022); Saharia et al. (2022); Rombach et al. (2022); Nichol et al. (2021); Ramesh et al. (2021) have showcased impressive capabilities in generating high-quality images guided by textual prompts. These models employ iterative denoising steps starting from a random noise map to learn the process of text-to-image generation. Latent Diffusion Model (LDM) Rombach et al. (2022) takes advantage of iterative denoising steps in a latent space, aiming to enhance text-to-image alignment and reduce training complexity while generating high-quality images from textual descriptions. Stable Diffusion and SDXL Podell et al. (2023) are applications of the Latent Diffusion method in text-to-image generation but trained with additional data and a powerful CLIP Radford et al. (2021) text encoder. DALL-E 2 Ramesh et al. (2021) and DALL-E 3 DAL (2023), state-of-the-art text-to-image generation model developed by OpenAI, achieve photorealistic T2I generation using diffusion-based models.

## 2.2 Controllable Image Generation

As text description cannot precisely control the position of generated instances, Some controllable text-to-image generation methods Gafni et al. (2022); Li et al. (2023); Avrahami et al. (2023); Bar-Tal et al. (2023); Zhou et al. (2024); Wang et al. (2024a); Zhang et al. (2023); Xie et al. (2023) introduce spatial conditioning controls to guide the image generation process. They extend the pre-trained T2I model Rombach et al. (2022) to integrate layout information into the generation and achieve control of instances' position. GLIGEN Li et al. (2023), MIGC Zhou et al. (2024), and InstanceDiffusion Wang et al. (2024a) are state-of-the-art methods which can support controlled image generation using discrete conditions such as bounding boxes. By integrating spatial conditioning controls, these methods enable users to have control over the positioning of instances in generated images. This advancement allows for fine-grained manipulation and customization in the image generation process.

## 2.3 LLM-assisted Image Generation

Large language models (LLMs) have transformed NLP tasks with their exceptional generalization abilities and applied to text-to-image generation. Methods like LLM-grounded DiffusionLian et al. (2023) and VPGenCho et al. (2024) use LLMs to determine object locations from text prompts via system prompts. LayoutGPTFeng et al. (2024a) enhances this by providing retrieved exemplars to the LLM. RanniFeng et al. (2024b) further incorporates a detailed semantic panel with multiple attributes to use LLMs' planning capabilities for painting tasks.

## 2.4 Knowledge Graph and LLM

**Knowledge Graph (KGs)** are structured multirelational knowledge bases that typically contain a set of facts. Each fact in a KG is stored in the form of triplet $(s, r, o)$, where $s$ and $o$ represent the subject and object entities, respectively, and $r$ denotes the relation connecting the subject and object entity. KGs are crucial for various applications as they offer accurate explicit knowledge Ji et al. (2021); Wang et al. (2023); Zhang et al. (2021); Sheu et al. (2021). **LLM**, pre-trained on the large-scale corpus, such as ChatGPT Brown et al. (2020) and GPT-4 OpenAI (2023) have showcased their remarkable capabilities in engaging in human-like communication and understanding complex queries, bringing a trend of incorporating LLMs in various fields Anil et al. (2023); Gunasekar et al. (2023); Jiang et al. (2023); Xue et al. (2023); Wang et al. (2024b); Chu et al. (2024a). By incorporating KGs, LLMs can benefit from the extensive knowledge stored in a structured and explicit manner. This integration enables LLMs to have a better understanding of the information contained in KGs, which also enhance the performance and interpretability of LLMs in various downstream tasks Pan et al. (2024). In our work, we leverage the knowledge retrieved from KGs to improve prompt analysis and object localization, enhancing the overall effectiveness of LLMs.

## 3 Method

In this section, we first provide a detailed definition of consistency hallucination in Sec 3.1. Following that, we delve into the details of our framework with object extraction and classification in Sec 3.2, relation extraction in Sec 3.3, object localization in Sec 3.4, and non-hallucinatory image generation in Sec 3.5. More details of our PCIG framework can be seen in Figure 2.

### 3.1 Consistency Hallucination Definition

Before delving into the methodology, it is essential to first define the types of hallucinations more detailed. Based on the MHaluBench Chen et al. (2024b) benchmark, our focus is centered upon four

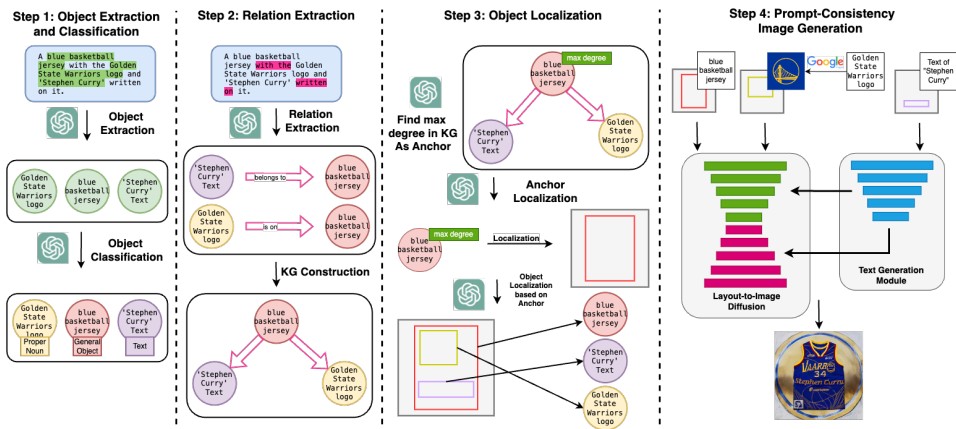

Figure 2: The pipeline of our PCIG method, using the example "A blue basketball jersey with the Golden State Warriors logo and 'Stephen Curry' written on it."

primary types of hallucinations that arise in text-to-image generation: (1) **AH(attribute hallucinations)**, where the attributes of objects in the generated images are incongruous with the provided prompts; (2) **OH(object hallucinations)**, where the number, placement, or other aspects of objects differ from the provided prompts; (3) **SCH(scene-text hallucinations)**, where the textual content within the generated images does not align with the given prompts;(4) **FH(factual hallucinations)**, where the depicted properties of objects contradict their real-world counterparts. While there are numerous other issues related to image hallucinations, this paper focuses chiefly on the aforementioned problems.

### 3.2 OBJECT EXTRACTION AND CLASSIFICATION

**Object Extraction.** The initial step of the method involves a meticulous process of identifying objects and their attributes from the textual prompt. Given the initial prompt $P$, we extract the objects present, their quantities, and their specific properties and structured them as $O = \{o_i\}_{i=1...N_o}$ where $o_i$ represents a concise caption of object, comprising the combination of attribute and object name(i.e. A grazing sheep), and $N_o$ denotes the number of the objects identified from the initial prompt. This step is imperative since understanding the precise object information is necessary to accurately generate an image that embodies these exact details. **Object Classification.** Following the object extraction, the next is to classify each identified object in $O$ into three specific categories $C = \{GO, TEXT, PN\}$ that $O = \{o_i, C\}_{i=1...N_o}$ where $GO, TEXT, PN$ represent general objects, text within the image, and objects that refer to proper nouns existing in the real world respectively. These categories are each linked to different types of hallucinations. $GO$ is associated with attribute and object hallucinations (AH and OH), $TEXT$ corresponds to scene-text hallucinations (SCH), and $PN$ is related to factual hallucinations (FH). This categorization is critical as it not only enables us to handle each hallucination problems separately for different types of objects but also lays the groundwork for subsequent relational and spatial analyses by clearly defining the nature and context of each object within the image.

### 3.3 RELATION EXTRACTION

**Relationship Recognition.** Once the objects are detected, GPT-4 determines the spatial relationships and interactions between the detected objects for the initial prompt. Let $R = \{r_i\}_{i=1...N_r}$ be the set of relationships identified from the initial prompt where $N_r$ is the number of relationships identified in the prompt. **Triple Generation.** Based on the detected objects $O$ and their relationships $R$, GPT-4 generates triples in the form of (object, predicate, object) to represent the identified relationships. The set of generated triples for the provided prompt is denoted as $T = \{t_i\}_{i=1...N_t}$, where $N_t$ is the number of triples in the initial prompt. Each triples is represented as $T = (O, R, O)$. For example, if the prompt depicts a young girl is wearing a pink dress, GPT-4 would generate a triple such as (Young Girl, is wearing, Pink Dress) where $\{YoungGirl, PinkDress\} \in O$ and $\{IsWearing\} \in R$. **Knowledge Graph Construction.** The knowledge graph the initial prompt is then constructed using the generated triples $T$ where the objects $O$ serve as the nodes and the relationships $R$ serve as the edges. The knowledge graph for the initial prompt is denoted as $G = (V, E)$ where $V = O$ is the set of nodes (objects) and $E$ is the set of edges with $R$ being the set of all pos-

| | Method | OH Acc.(%) | TH Acc.(%) | FH Acc.(%) | TFH Acc.(%) | Overall Acc.(%) |
|---|---|---|---|---|---|---|
| Text-to-Image | SDv1.6 Rombach et al. (2022) | 15.33 | 11.11 | 22.22 | 0.00 | 14.55 |
| | SDXL Podell et al. (2023) | 18.98 | 9.52 | 8.33 | 0.00 | 15.91 |
| | DALL-E 2 Ramesh et al. (2021) | 24.82 | 7.94 | 0.00 | 0.00 | 17.73 |
| | DALL-E 3 DAL (2023) | 60.58 | 26.98 | 9.99 | 0.00 | 45.45 |
| Layout-to-Image | GLIGEN Li et al. (2023) | 88.32 | 7.94 | 22.22 | 0.00 | 59.09 |
| | MIGC Zhou et al. (2024) | 94.16 | 11.11 | 8.33 | 0.00 | 63.18 |
| | InstanceDiffusion Wang et al. (2024a) | 95.62 | 9.52 | 22.22 | 0.00 | 64.09 |
| LLM-Assisted | LayoutGPT Feng et al. (2024a) | 88.32 | 7.94 | 22.22 | 0.00 | 59.09 |
| | LayoutLLM-T2I Qu et al. (2023) | 94.16 | 11.11 | 8.33 | 0.00 | 63.18 |
| | VPGen Cho et al. (2024) | 95.62 | 9.52 | 22.22 | 0.00 | 64.09 |
| | Ranni Feng et al. (2024b) | 95.62 | 9.52 | 22.22 | 0.00 | 64.09 |
| | LMD Lian et al. (2023) | 95.62 | 9.52 | 22.22 | 0.00 | 64.09 |
| | **PCIG (ours)** | **94.89** | **82.54** | **77.78** | **50.00** | **89.55** |

Table 1: Experimental results of our framework and various baseline on MHaluBench dataset.

sible relationships in the initial prompt. The construction of the knowledge graph using GPT-4 is a critical step in our method. It provides a structured and detailed representation of the initial prompt, capturing the relationships and interactions between objects in a way that goes beyond simple semantic features Chu et al. (2024c). This enriched representation proves advantageous for subsequent steps of object localization and image generation.

### 3.4 OBJECT LOCALIZATION

**Spatial Organization.** Building upon the relationship extraction, this part focuses on the spatial organization of objects within the canvas. The process begins by identifying the node with the max degree $V_m$ in the knowledge graph $G$ constructed previously, highlighting it as the pivotal object in the image composition. Determining this anchor object is critical for orienting other objects in relation to it, ensuring a coherent and realistic spatial arrangement. **Bounding Box Generation.** The meticulous spatial arrangement extends to the precise calculation of each object's placement in relation to the anchor point and throughout the canvas expanse. This phase demands a fine-tuned equilibrium to instill visual authenticity, taking into account factors like object scale and spatial positioning to construct bounding boxes that convincingly outline the locations of the entities. Let $BB = \{[x_i, y_i, w_i, h_i]\}_{i=1...N_o}$ denote the set of bounding boxes for the objects, where the tuple $(x, y)$ signifies the object's coordinates on the canvas, and $(w, h)$ indicates the object's dimensions within the space. The bounding boxes are precisely structured, conforming to exact dimensional specifications and coordinate precisions, ensuring that every object is proportionately and accurately depicted within the generated image.

### 3.5 PROMPT-CONSISTENCY IMAGE GENERATION

Building upon the previously described steps, we have successfully secured a series of well-defined bounding boxes for every object on the canvas. Our objective is to leverage these bounding boxes to generate corresponding images wherein the positioning of objects closely mirrors the layout specified by $BB$. To this end, we employ a controllable text-to-image model as our primary framework of the model that is specifically designed to accept bounding boxes as input, enabling precise manipulation of image outcomes. A visual text generation module is incorporated with the model, designated to handle linguistic elements, forming the essence of our integrated system.

Our system categorizes inputs into three segments as mentioned in Sec. 3.2: GO, TEXT, and PN for image generation. For GO, we input both the bounding box and its caption directly into the main framework of the model. For TEXT, we input the textual content and its corresponding bounding box into a visual text generation module. This module incorporates narrative elements into the visual output. For PN, we use a search engine to find representative images of the objects. These images, along with their bounding boxes, are seamlessly integrated into the model's primary input stream. By categorizing objects and applying specific generation paradigms, our model prevents the generation of images with hallucination features. This methodical approach results in photorealistic and prompt-consistency images, eliminating the challenges posed by hallucinations.

## 4 EXPERIMENTS
### 4.1 EXPERIMENTS SETTINGS
**Settings for MHaluBench.** MHaluBench Chen et al. (2024b) is a benchmark which encompasses the content from text-to-image generation, aiming to rigorously assess the advancements in multimodal hallucination detectors. The benchmark has been meticulously curated to include

| Method | spatial | shape | color | counting | texture | other |
|---|---|---|---|---|---|---|
| SDv1.6 Rombach et al. (2022) | 13.12 | 36.46 | 37.30 | 18.01 | 42.19 | 22.55 |
| SDXL Podell et al. (2023) | 20.86 | 47.80 | 60.50 | 19.92 | 54.46 | 28.05 |
| LayoutGPT Feng et al. (2024a) | 35.13 | 37.67 | 38.00 | 31.64 | 42.33 | 32.14 |
| LayoutLLM-T2I Qu et al. (2023) | 28.24 | 33.17 | 36.30 | 24.54 | 43.33 | 34.97 |
| VPGen Cho et al. (2024) | 29.03 | 48.67 | 60.02 | 29.42 | 54.00 | 36.57 |
| Ranni Feng et al. (2024b) | 31.67 | 49.34 | 68.93 | 27.20 | 63.25 | 44.03 |
| LMD Lian et al. (2023) | 27.04 | 54.62 | 54.95 | 27.33 | 52.41 | 35.47 |
| **PCIG (ours)** | **72.79** | **65.82** | **88.73** | **74.75** | **79.67** | **43.22** |

Table 2: Experimental results of our framework and various baseline on T2I-CompBench dataset.

| Method | OH Acc.(%) | TH Acc.(%) | FH Acc.(%) | TFH Acc.(%) | Overall Acc.(%) |
|---|---|---|---|---|---|
| w/o KG extraction | 64.96 | 82.54 | 77.78 | 0.00 | 70.45 |
| w/o Object extraction | 75.91 | 7.94 | 11.11 | 0.00 | 50.45 |
| w/o Text module | 95.62 | 9.52 | 22.22 | 0.00 | 64.09 |
| **model (ours)** | **94.89** | **82.54** | **77.78** | **50.00** | **89.55** |

Table 3: Ablation results of our PCIG method on MHaluBench dataset.

220 exemplars dedicated to Text-to-Image Generation with 158 are hallucinatory and 62 are non-hallucinatory. Specifically, it includes 137 prompts which will generate images with object and attribute hallucination potentially, 63 prompts with textual hallucination, 18 prompts with factual hallucination, 2 prompts with combination of factual hallucination and textual hallucination. **Baseline.** Our baseline divides into three parts. The first part is the comparison with the most representative generative models, including Stable Diffusion v1.6 Rombach et al. (2022), SDXL Podell et al. (2023), DALL-E 2 Ramesh et al. (2021), and DALL-E 3 DAL (2023), which generate visually detailed images directly based on the prompt in the benchmark. The second part is the comparison with the state-of-the-art controllable text-to-image models, also named layout-to-image models, including GLIGEN Li et al. (2023), MIGC Zhou et al. (2024), and InstanceDiffusion Wang et al. (2024a), which introduce spatial conditioning controls to guide the image generation process. We will use the bounding box generated in the first three steps to guide the process of image generation. The last part is the comparison with the LLM-assisted image generation methods including Layout-GPTFeng et al. (2024a), LayoutLLM-T2IQu et al. (2023), VPGenCho et al. (2024), RanniFeng et al. (2024b), and LMDLian et al. (2023). **Metric.** We utilize UNIHD scoreChen et al. (2024b), which will return a label represents whether the input image with corresponding prompt is hallucinatory or not,as our hallucination detection method for generated images. We calculate the accuracy for each hallucination type mentioned above, including object hallucination accuracy (OH acc.), textual hallucination accuracy (TH acc.), factual hallucination accuracy (FH acc.), textual and factual hallucination accuracy (TFH acc.), and overall accuracy for evaluation.

**Settings for T2I-CompBench.** T2I-CompBench Huang et al. (2023) is a comprehensive benchmark for open-world compositional text-to-image generation, consisting of 6,000 compositional text prompts from 3 categories (attribute binding, object relationships, and complex compositions) and 6 sub-categories (color binding, shape binding, texture binding, spatial relationships, non-spatial relationships, and complex compositions). **Baseline.** we compare our method with Stable Diffusion v1.6 Rombach et al. (2022), SDXL Podell et al. (2023), LayoutGPTFeng et al. (2024a), LayoutLLM-T2IQu et al. (2023), VPGenCho et al. (2024), RanniFeng et al. (2024b), and LMDLian et al. (2023) to further demonstrate the efficacy of our approach. **Metric.** We use BLIP-VQA scoreLi et al. (2022) for color, shape, and texture tasks as well as UniDet scoreZhou et al. (2022) for spatial, counting and other (non-spatial and complex) tasks.

**Implement Details** Our pipeline is training-free and comprises three pre-trained models. We employ the GPT-4 OpenAI (2023) as the base LLMs to generate bounding box for identified objects and choose InstanceDiffusion Wang et al. (2024a) as primary controllable text-to-image model while AnyText Tuo et al. (2023) as text-generation module.

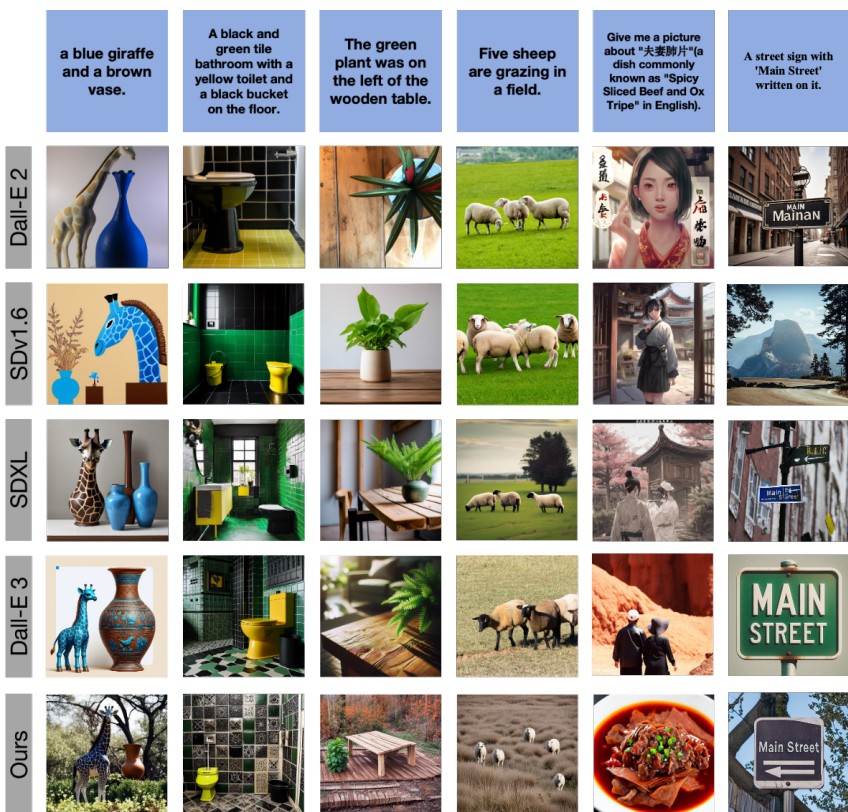

Figure 3: Compared with multiple text-to-image generation methods. Our method shows comparable performance in all aspects.

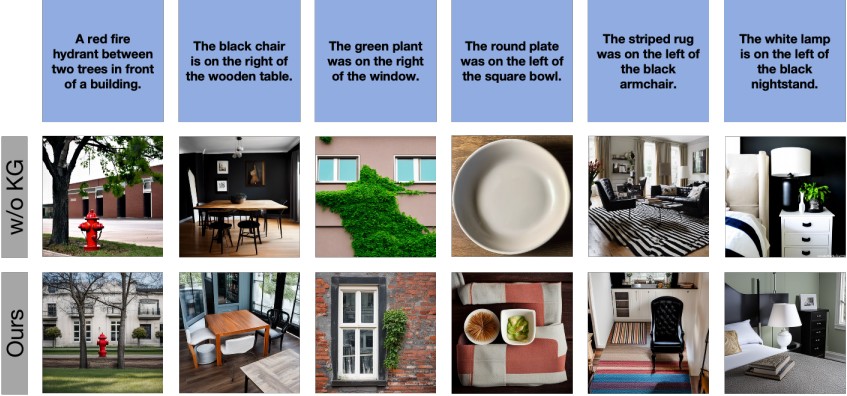

Figure 4: Ablation study on knowledge graph construction. Results become inaccurate in object locations when the proposed module is disable.

## 4.2 EXPERIMENTAL RESULTS AND QUALITATIVE ANALYSIS

**Experimental Results.** Table 1 shows that PCIG outperforms the baseline models in all metrics on MHaluBench dataset. It is worth noting that the object hallucination accuracy of all text-to-image models, especially in Stable Diffusion Rombach et al. (2022) and DALL-E 2 Ramesh et al. (2021), is extremely low. This suggests that these models struggle to generate images that align with the given prompts under such conditions. On the other hand, PCIG and other competitive layout-to-image models demonstrate exceptional abilities in accurately generating objects and their attributes in image generation. In terms of text hallucination accuracy, factual hallucination accuracy, and textual and factual hallucination accuracy, all baseline models perform poorly. In contrast, PCIG stands out from the rest. With the help of prompt analysis and text generation module, PCIG showcases exceptional performance in both text hallucination accuracy and factual hallucination accuracy. It

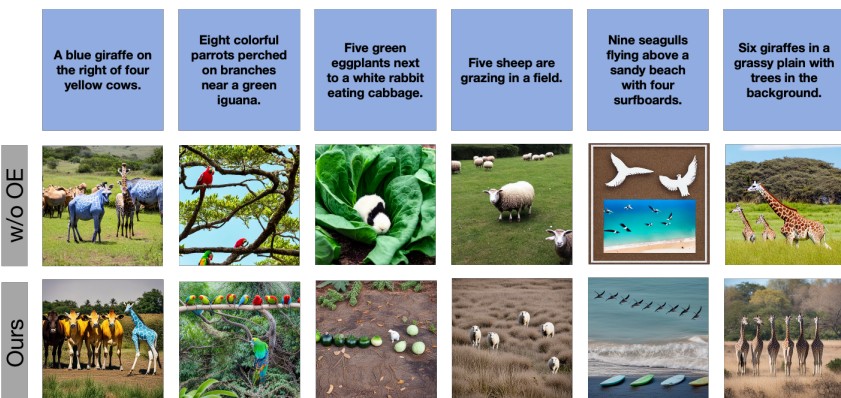

Figure 5: Ablation study on object extraction. Results become inaccurate in object count and attribute when the proposed module is disable.

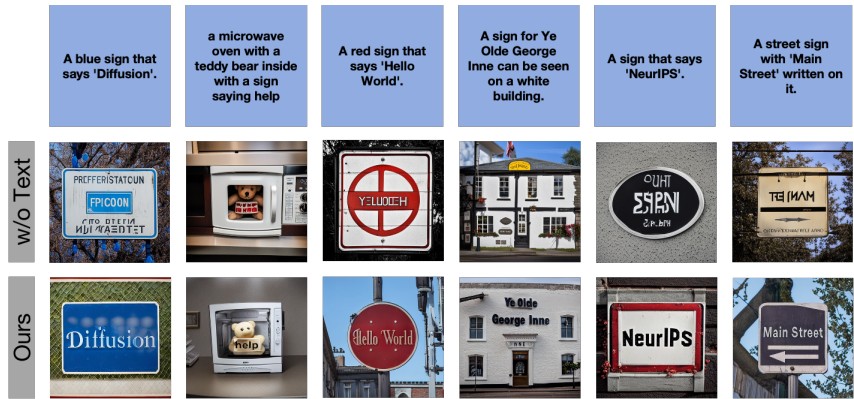

Figure 6: Ablation study on text generation module. Results become inaccurate in visual text when the proposed module is disable.

surpasses the baseline models, highlighting its impressive capabilities in text generation and factual object generation. The corresponding prompt template is shown in Figure 9

As shown in the Table 2, our PCIG method significantly outperforms existing LLM-assisted approaches across all attributes. **Attribute accuracy.** While methods like LayoutGPTFeng et al. (2024a) and LayoutLLM-T2IQu et al. (2023) struggle with attributes such as color (0.3800 and 0.3630 on T2I-CompBench Huang et al. (2023)) and texture (0.4233 and 0.4333 on T2I-CompBench Huang et al. (2023)), our method achieves much higher accuracy (0.9385 for color and 0.7967 for texture on T2I-CompBench Huang et al. (2023)). This is because these methods primarily focus on layout generation without considering specific attributes like color, size, material, and shape. **Spatial reasoning.** Our method excels in spatial accuracy (0.7279) compared to others (e.g., LayoutGPTFeng et al. (2024a) at 0.3513). This is due to our innovative approach of using relationship extraction and knowledge graphs to generate more accurate and proportionate bounding boxes. While other methods consistently produce small bounding boxes(e.g., 80x20 for a person) that fail to utilize the full canvas regardless of the scene complexity, our method dynamically adjusts bounding box sizes based on object relationships and quantity. For instance, in scenes with fewer objects, our method might generate a person's bounding box as 200x100, fully utilizing the canvas. In more complex scenes with multiple objects, the bounding boxes appropriately scale down to accommodate all elements. This adaptive spatial representation allows methods like InstanceDiffusion to generate clearer and more accurate images. **Counting.** Our method shows substantial improvements in counting accuracy (0.7475 compared to the next best of 0.3164). In conclusion, these demonstrate the effectiveness of our comprehensive approach in handling various aspects of image generation consistently. The detailed analysis for extracting objects, identifying text, entities and relation are in Appendix A.3.

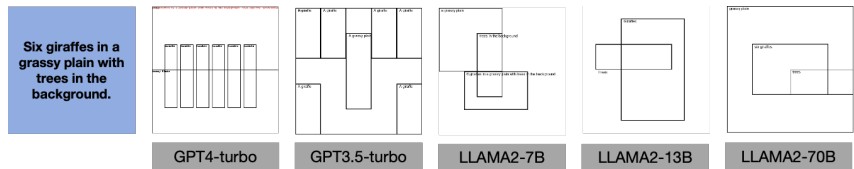

Figure 7: Bounding box generated by different LLM with original prompt "Six giraffes in a grassy plain with trees in the background.".

| model | GLIGEN Li et al. (2023) | | MIGC Zhou et al. (2024) | | InstanceDiffusion Wang et al. (2024a) | |
|---|---|---|---|---|---|---|
| | OH Acc.(%) | TH Acc.(%) | OH Acc.(%) | TH Acc.(%) | OH Acc.(%) | TH Acc.(%) |
| Base. | 88.32 | 7.94 | 94.16 | 11.11 | 95.62 | 9.52 |
| Base. w/ text module | 89.05 | 76.19 | 94.16 | 79.37 | 94.89 | 82.54 |
| Δ | **+0.73** | **+68.25** | **+0.00** | **+68.25** | **-0.87** | **+73.02** |

Table 4: Comparison results of different base controllable text-to-image model with and without text module on MHaluBench dataset.

**Qualitative Analysis.** Through visualization of generated images, we compare our PCIG with competitive text-to-image generation models (SD Rombach et al. (2022), SDXL Podell et al. (2023), DALL-E 2 Ramesh et al. (2021), and DALL-E 3 DAL (2023)). As depicted in Figure 3, The first row represents the prompts given to generate images and the left column represents the model used to generate images based on prompts. As a result, Stable Diffusion, SDXL, DALL-E 2, and DALL-E 3 shows different types of visualization errors during generation, including the inconsistency between the prompts and object attributes in generated images (column 1 and 2), the inconsistency between the prompts and object locations in generated images (column 3), the inconsistency between the prompts and the number of objects in generated images (column 4), text generation error (column 5), and factual object generation error (column 6). In contrast, Leveraging the capabilities of prompt analysis and text generation module, our PCIG presents accurate and vivid images consistent with the original prompts, as shown in the last row.

### 4.3 ABLATION STUDY

**w/o KG extraction.** For w/o KG extraction, the ablation experiment on the MHaluBench dataset locates identified objects without relation extraction and knowledge graph construction, lacking relation and spatial analysis for prompts. The prompt template is shown in Figure 10. Table 3 reveals that without a knowledge graph, the model struggles to fully comprehend relationships between identified objects, resulting in inaccurate localization. Figure 4 compares our method with the ablation results. The experiments illustrate issues arising from the absence of objects in the prompt (column 1 and 4) and inaccuracies in positional relationships (rest of column). These findings emphasize the importance of relation extraction and knowledge graph construction in prompt analysis.

**w/o object extraction.** For w/o object extraction, the ablation experiment on MHaluBench dataset focuses on extracting relationships between objects without considering specific object information. The corresponding prompt template is shown in Figure 11. Table 3 clearly demonstrates that when the model lacks object information, it faces challenges in accurately identifying object attributes and the number of objects while extracting relationships between them. Furthermore, it also struggles in correctly identifying object categories when generating textual and factual object. Figure 5 presents a comparison between our method and the ablation results. The ablation experiment vividly highlights the problem of inconsistency between the number of objects in the generated image and the expected number of objects mentioned in the original prompt. Consequently, the model fails to provide precise object number and attribute information due to the absence of object guidance. which prove the importance of object extraction in prompt analysis.

**w/o text module.** For w/o text module, the ablation experiment on MHaluBench dataset aimed to examine the impact of removing the text generation module in our model. The results, shown in Table 3, highlight that without the text generation module, the model faced challenges in generating accurate text. Figure 6 provides a visual comparison between our method and the ablation results. The ablation experiments demonstrate that errors, such as missing and incorrect text, were prevalent

| Model | GPT4-turbo OpenAI (2023) | LLAMA2-7B Touvron et al. (2023) | LLAMA2-13B Touvron et al. (2023) | LLAMA2-70B Touvron et al. (2023) | GPT3.5-turbo Brown et al. (2020) |
|---|---|---|---|---|---|
| Overall Acc.(%) | 89.54 | 32.27 | 42.27 | 67.27 | 70.91 |
| Δ | +0.00 | -57.27 | -47.27 | -22.27 | -18.64 |

Table 5: Comparison results of different LLM for our PCIG method on MHaluBench dataset.

| Model | SDXLPodell et al. (2023) | DALL-E 3DAL (2023) | LayoutGPTFeng et al. (2024a) | LayoutLLM-T2IQu et al. (2023) | VPGenCho et al. (2024) | RanniFeng et al. (2024b) | LMDLian et al. (2023) | PCIG(ours) |
|---|---|---|---|---|---|---|---|---|
| Inference time(s) | 8.77 | 14.20 | 35.48 | 34.94 | 37.31 | 26.62 | 31.35 | 33.43 |

Table 6: Inference time for generating image "A living room and dining room have two tables, one couche, and three chairs. high quality. professional photo." with various methods.

without the text generation module. These findings reinforce the significance of the text generation module in our approach.

**Different base controllable text-to-image models.** In this section, we conducted ablation experiments using different baseline controllable text-to-image generation models, including GLIGEN Li et al. (2023), MIGC Zhou et al. (2024), and InstanceDiffusion Wang et al. (2024a), with and without a text generation module. Table 4 displays the outcomes of ablation experiments conducted on various controllable T2I models, focusing on object hallucination accuracy and text hallucination accuracy. The findings reveal that utilizing different models, with or without a text generation module, both yields outstanding results for object hallucination accuracy. Furthermore, the presence of a text generation module significantly enhances text hallucination accuracy. This implies that the text generation module can be seamlessly integrated into different base models to improve text generation capabilities.

**Different LLM for prompt analysis.** In this ablation experiment, we test the performance of different language models (LLMs) for prompt analysis. The LLMs we used are GPT4-turbo OpenAI (2023), GPT3.5-turbo Brown et al. (2020), LLAMA2-7B Touvron et al. (2023), LLAMA2-13B, and LLAMA2-70B. We measure the overall accuracies of these models, and the results are summarized in Table 5. According to the results, GPT4-turbo demonstrats the highest level of competitiveness among the LLMs tested. On the other hand, LLAMA2-7B performs the least effectively compared to the other models. Figure 7 displays the bounding boxes generated by different language models when analyzing the prompt "Six giraffes in a grassy plain with trees in the background". GPT4-turbo accurately identifies all objects and provides reasonable positions. GPT3.5-turbo can identify all objects, but the positions it generates are unreasonable. All LLAMA model fail to recognize objects and also generate unreasonable positions.

**Inference time for different image generation methods.** In this section, we analyzed the inference time for various methods. Table 6 presents the inference time for generating the image description: "A living room and dining room have two tables, one couch, and three chairs. High quality. Professional photo." using different image generation methods and several LLM-assisted image generation methods. Generally, traditional image generation methods are faster than LLM-assisted methods. The inference times for LLM-assisted methods are similar, as they all incorporate LLMs, which increases the inference time.

## 5 CONCLUSION

In this paper, we introduced the **P**rompt-**C**onsistency **I**mage **G**eneration(PCIG), a effective approach that significantly enhances the alignment of generated images with their corresponding descriptions. Leveraging a state-of-the-art large language module, we make a comprehensive prompt analysis and generate bounding box for each identified objects. We further integrate a state-of-the-art controllable image generation model with a visual text generation module to generate an image guided by bounding box. We demonstrate our method could handle various type of object category based on the integration of text generation module and search engine. Both qualitative and quantitative results demonstrate our superior performance.

**Limitations.** Our method uses GPT4-turbo as our LLM to finish object extraction, relation extraction, and object localization, which costs approximately 0.08$ in one generation process. Furthermore, our method have difficulties in generating images with complex relationship and interaction between objects as well as with small text. To address this concern, A more powerful basic diffusion model would be of great help. More discussion for failure cases are in Appendix A.2

## 6 *

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

## A  APPENDIX / SUPPLEMENTAL MATERIAL

### A.1  BACKGROUND AND RELATED WORK

**Visual Text Generation.**   Current mainstream text-to-image generation models, like Stable Diffusion, excel at producing high-quality images. However, they struggle to generate accurate and legible text on these images. To address this limitation, recent research studies Ma et al. (2023); Yang et al. (2024); Tuo et al. (2023); Cao et al. (2024); Chen et al. (2024a); Chu et al. (2024b) have focused on integrating clear and readable text into images by introducing glyph conditions in the latent space. These advancements, particularly, GlyphControl Yang et al. (2024) and AnyText Tuo et al. (2023), can be seamlessly plugged into existing diffusion models, allowing for more precise rendering of text on generated images.

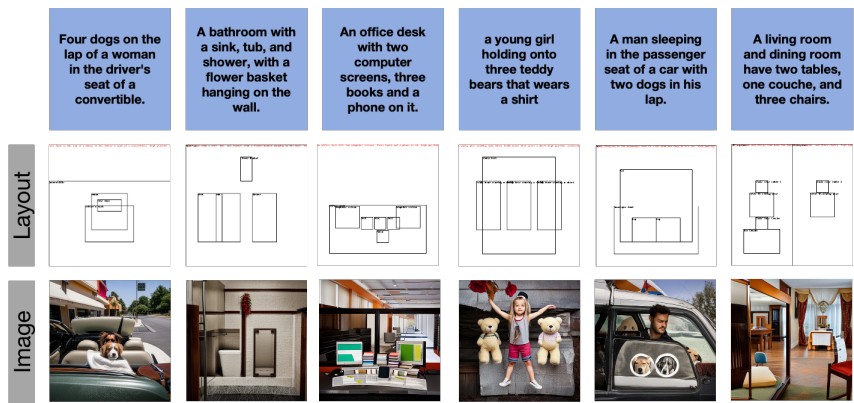

Figure 8: Failure cases of our PCIG method on MHaluBench.

### A.2  FAILURE CASES

Figure 8 shows several failure cases of our PCIG methid. We attribute the failure cases to the following three issues: **Generated images fail to depict complex physical interactions**, which is due to limitations in the capabilities of the controllable generation model. For example, the model may struggle with prompts that describe intricate physical processes or dynamic scenes with multiple interacting elements. **Generated images fail to accurately represent complex spatial layouts**, which is also a result of limitations in the controllable generation model's capabilities. Prompts requiring very intricate arrangements of multiple objects or elaborate scenes with specific spatial configurations may exceed the model's current abilities.**Failure to extract objects or object attributes**, especially for special objects like text, which is caused by errors in the object extraction step. This could occur with prompts containing unusual or abstract concepts, complex metaphors, or highly technical terminology, leading to missing or incorrectly identified elements in the final image.

## A.3   LLM OUTPUT ANALYSIS

The accuracy of LLM output is very important as the proposed method relies on GPT-4 to generate objects, entities, relation and text. In this section, We demonstrate the accuracy of our LLM outputs on MHaluBench dataset from multiple perspectives in different stages. All ground truth values for the number of objects, object types, object descriptions, and relation extraction in prompts are human-annotated.

**Extracting Objects.**   We evaluate LLM's accuracy in extracting objects from prompts. The results show a high precision of 0.9598, a lower recall of 0.8396, and an F1 score of 0.8956 for accuracy in extracting objects from prompts. The high precision indicates that when objects are identified, they are usually correct. However, the lower recall suggests that some objects are being missed. This discrepancy can be attributed to prompts containing objects with unspecified quantities. In ideal scenarios, the model should generate specific numbers for these objects. However, it sometimes treats a group of objects with an uncertain quantity as a single entity.

**Generating Captions.**   We evaluate LLM's accuracy in generating correct captions for every objects. The accuracy approaches 1.0000, indicating that nearly all extracted captions are correct. Exceptions occur primarily when dealing with unusual or abstract concepts and text extraction.

**Classifying Objects.**   We evaluate LLM's accuracy in classifying objects into different categories. The accuracy is 1.0000, which means all classification is correct.

**Extracting Relationships.**   We evaluate LLM's ability to extract relationships between objects. The results for accuracy in extracting relationships between objects show a precision of 0.7763, a high recall of 0.9817, and an F1 score of 0.8670. The lower precision coupled with the very high recall indicates that the model is highly effective at identifying most relevant relationships but tends to overgenerate. This overgeneration manifests in two main ways. Firstly, the model sometimes produces extraneous relationships that aren't explicitly stated or necessarily implied in the prompt. Secondly, it occasionally generates relationships involving background objects that, while potentially correct in the context of the image, aren't directly related to the primary objects extracted from the prompt.

## A.4   COMPLETED PROMPT

In this section, we first outline the prompt template in Figure 9 designed to guide the object extraction, relation extraction, and object localization. Then we present the prompt template designed for ablation study in Figure 10 and Figure 11. Furthermore, we present more results of our PCIG method in Figure 12.

**Give a prompt, Imagine there is an image captioned by the prompt, perform the following actions:**
**- Identify as many objects and their corresponding attributes(###COLOR, ACTION.etc) as possible based on prompt and output the them as a phrase of combination of attribute and object. ###Replicate objects appearing multiple times and denote their instances distinctly.**
**- Classify the objects into specific category: 1.general objects. 2.text in the image. 3.Objects that specifically refer to the Proper Noun of the real world(###One entity only be classified in one category)**
**- Identify as many relations among objects in Step 2 as possible and output a list in the format [ENTITY 1, TYPE of ENTITY 1, RELATION, ENTITY 2, TYPE of ENTITY 2] and the type should be the objects' category.**
**- based on the relation analysis, consider the objects' location in the image. follow the three step below: 1. Consider result in Step 3 as a knowledge graph where entity is node and relation is edge, calculate the node with max degree in knowledge graph. 2. Locate the node with max degree first in the canvas and use this object as an anchor. 3. Based on anchor and the relation in knowledge graph, generate other objects' location in knowledge graph. Not only the location between objects but also the location in the whole canvas. Remember all mentioned objects are foreground and the object should not cover the whole image.**
**- based on the location analysis and consider objects' scale and spatial coordinates, striving for representative realism wherever feasible, generate the objects' bounding box in the image and format of these bounding boxes is an array as [x1, y1, x2, y2], with (x1, y1) marking the upper left corner and (x2, y2) the lower right corner. Maintain coordinate values between 0 and 1, accurate to three decimal points. the width and height of bounding box should be bigger than 0.105 in min and smaller than 0.895 in max.**
**- Convert the result of Step 5 strictly adhering to the following structure:**
**{{"object1": {{"phrase": "the name of object","coordinates": [x1, y1, x2, y2]}},"object2": {{"phrase": "the name of object","coordinates": [x1, y1, x2, y2]}},...}}**

**Here are some examples:**
**...**

**prompt:**
**output:**

Figure 9: Prompt template of prompt analysis in PCIG method.

**Give a prompt, Imagine there is an image captioned by the prompt, perform the following actions:**

**- Identify as many objects and their corresponding attributes(###COLOR, ACTION.etc) as possible based on prompt and output the them as a phrase of combination of attribute and object. ###Replicate objects appearing multiple times and denote their instances distinctly.**

**- Classify the objects into specific category: 1.general objects. 2.text in the image. 3.Objects that specifically refer to the Proper Noun of the real world(###One entity only be classified in one category)**

**- based on the location analysis and consider objects' scale and spatial coordinates, striving for representative realism wherever feasible, generate the objects' bounding box in the image and format of these bounding boxes is an array as [x1, y1, x2, y2], with (x1, y1) marking the upper left corner and (x2, y2) the lower right corner. Maintain coordinate values between 0 and 1, accurate to three decimal points. the width and height of bounding box should be bigger than 0.105 in min and smaller than 0.895 in max.**

**- Convert the result of last Step strictly adhering to the following structure: {{"object1": {{"phrase": "the name of object","coordinates": [x1, y1, x2, y2]}},"object2": {{"phrase": "the name of object","coordinates": [x1, y1, x2, y2]}},...}}**

**prompt:**

**output:**

Figure 10: Prompt template of ablation study on knowledge graph construction in PCIG method.

**Give a prompt, Imagine there is an image captioned by the prompt, perform the following actions:**
**- Identify as many relations in prompt as possible and output a list in the format [ENTITY 1, TYPE of ENTITY 1, RELATION, ENTITY 2, TYPE of ENTITY 2] and the type should be the objects' category. multiple ENTITY MUST be a group.**
**- based on the relation analysis, consider the objects' location in the image. follow the three step below: 1. Consider result in last Step as a knowledge graph where entity is node and relation is edge, calculate the node with max degree in knowledge graph. 2. Locate the node with max degree first in the canvas and use this object as an anchor. 3. Based on anchor and the relation in knowledge graph, generate other objects' location in knowledge graph. Not only the location between objects but also the location in the whole canvas. Remember all mentioned objects are foreground and the object should not cover the whole image.**
**- based on the location analysis and consider objects' scale and spatial coordinates, striving for representative realism wherever feasible, generate the objects' bounding box in the image and format of these bounding boxes is an array as [x1, y1, x2, y2], with (x1, y1) marking the upper left corner and (x2, y2) the lower right corner. Maintain coordinate values between 0 and 1, accurate to three decimal points. the width and height of bounding box should be bigger than 0.105 in min and smaller than 0.895 in max.**
**- Convert the result of last Step strictly adhering to the following structure: {{"object1": {{"phrase": "the name of object","coordinates": [x1, y1, x2, y2]}},"object2": {{"phrase": "the name of object","coordinates": [x1, y1, x2, y2]}},...}}**
**prompt:**
**output:**

Figure 11: Prompt template of ablation study on object extraction in PCIG method.

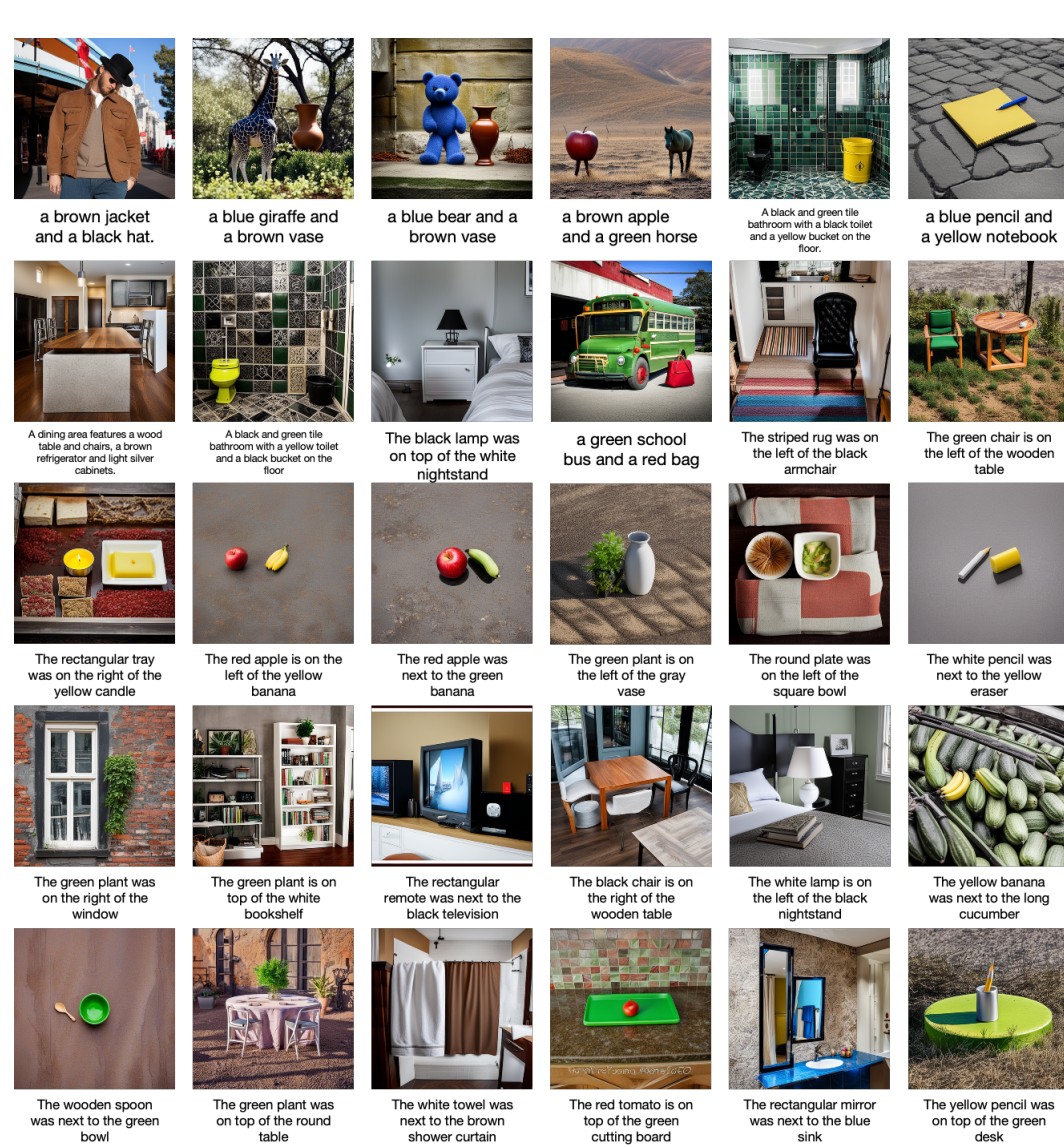

Figure 12: More results of our PCIG method.

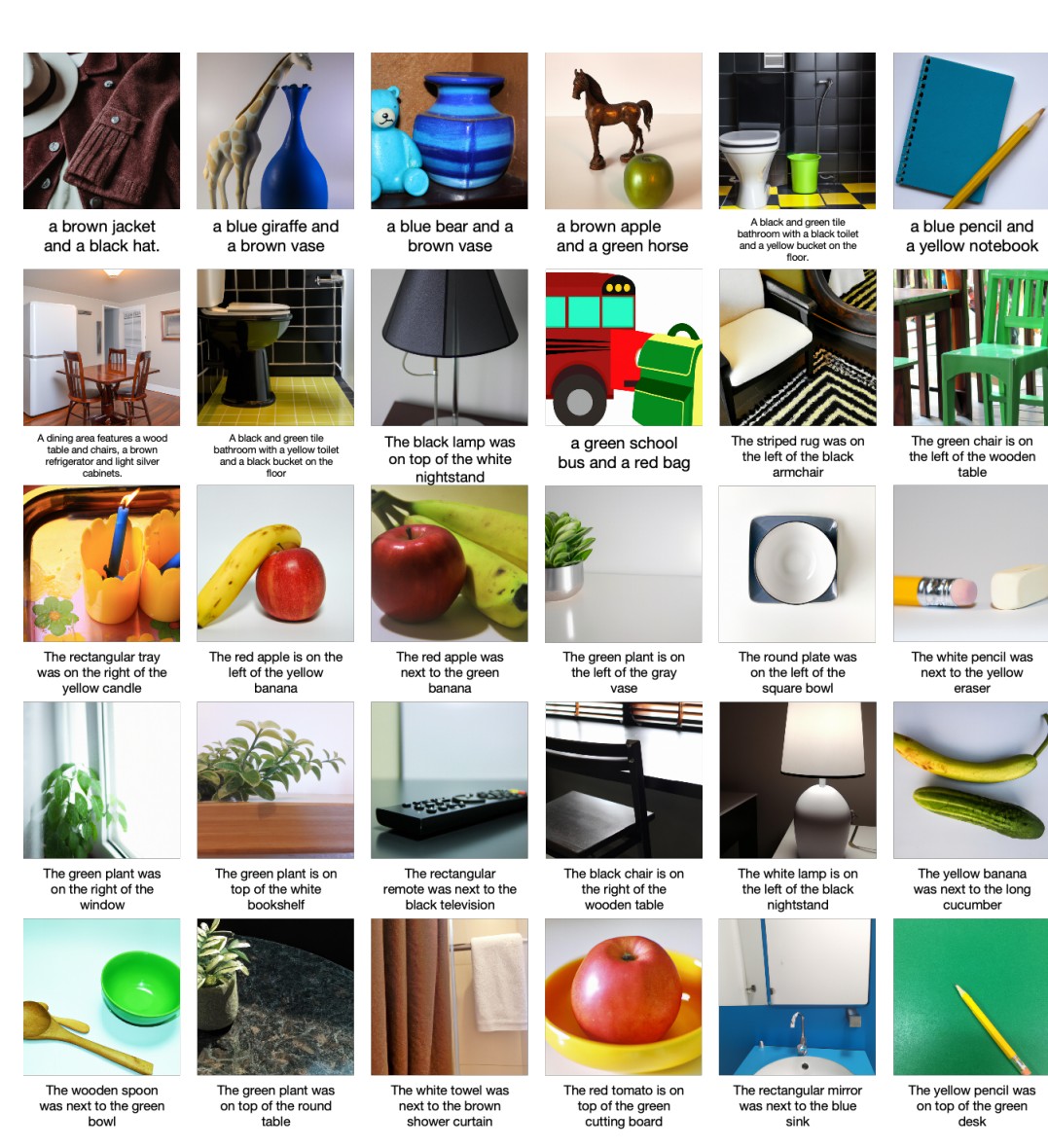

Figure 13: More hallucination results of DALL-E 2.

