# OpenReview forum: "Prompt-Consistency Image Generation (PCIG): A Unified Framework Integrating LLMs, Knowledge Graphs, and Controllable Diffusion Models"
_ICLR.cc/2025/Conference — ICLR 2025 Conference Withdrawn Submission_

### Official Review · Reviewer_ReQL · 2024-10-22

**Soundness:** 3
**Presentation:** 3
**Contribution:** 2
**Rating:** 5
**Confidence:** 4

**Summary:**

The paper introduces PCIG, a novel framework aimed at addressing inconsistencies in text-to-image generation. By integrating Large Language Models (LLMs), knowledge graphs, and controllable diffusion models, the proposed method enhances the alignment between generated images and their corresponding text prompts. The framework handles three key aspects of consistency: general objects, text within images, and objects that refer to proper nouns existing in the real world. Through extensive experimentation on the MHaluBench dataset, PCIG demonstrates significant improvements in both quantitive and qualtitive evaluations.

**Strengths:**

The modular design of the framework shows potential for scalability and adaptability to more complex scenarios.

Consistency between the text prompt and the generated image is a well-recognized challenge in text-to-image generation. Solving this issue has significant implications for advancing practical applications of AI-generated content, making the PCIG framework highly valuable.

The paper presents extensive quantitative and qualitative experiments, as well as ablation studies, that clearly demonstrate PCIG’s prompt consistency.

**Weaknesses:**

The primary limitation of this work lies in its novelty. PCIG appears to be a straightforward combination of pre-existing models and methods, offering little in terms of new insights or theoretical advancements. While the integration is effective, the framework lacks a significant innovative contribution that differentiates it from prior work.

The demo presented in Figure 2 still exhibits inconsistencies. For instance, the team name "Warriors" is misspelled as "VAARR," and Stephen Curry’s jersey number, which should be 30, is incorrect. These errors detract from the credibility of the framework’s performance in ensuring image-prompt consistency.

The experimental section lacks widely accepted image generation evaluation metrics, such as the Frechet Inception Distance (FID) and Inception Score (IS). Including these metrics would have provided a more comprehensive evaluation of the model's performance and allowed for better comparison with existing state-of-the-art methods.

Some experimental results, particularly in Table 3, raise concerns. For example, the OH ACC in the first column is unexpectedly higher when the text module is omitted. Moreover, the remove of KG extraction does not seem to improve TH or FH ACC, calling into question the contribution of these modules.

**Questions:**

Please see weakness.

---

### Official Review · Reviewer_cvFW · 2024-10-24

**Soundness:** 2
**Presentation:** 1
**Contribution:** 1
**Rating:** 3
**Confidence:** 4

**Summary:**

The paper presents a framework (PCIG) tackling the issue of text-to-image (T2I) generation inconsistencies. It highlights important challenges such as entity mismatches and handling complex scenes, and it proposes a pipeline that integrates large language models (LLMs), knowledge graphs and location prediction to improve overall generation quality.

**Strengths:**

1. The authors have summarized some of the key challenges in text-driven image generation, particularly around maintaining consistency in entities (Figure 1) and managing complex scene generation.
2. The proposed pipeline is relatively complete and utilizes LLM to refine entities and layouts, which helps improve text-consistent image generation.

**Weaknesses:**

I have some concerns about different aspects of the paper that I feel should be addressed:

1. **Framework Innovation**: Actually, what concerns me the most is that the proposed approach feels more like a combination of several established LLM strategies, such as “object extraction,” “relation extraction,” and “object localization,” rather than introducing novel research contributions. This makes it come across more as an industrial solution, which might not fit the innovation threshold for a conference like ICLR.

2. **Quantitative Results**: I find it difficult to be fully convinced by the quantitative results. The use of the existing *AnyText* [1] module to mitigate textual hallucination and factual hallucination (improved *TH Acc* and *FH acc*) seems to remove much of the novelty from your approach, as seen in Tables 1, 3.

3. **Misleading References**:
In my view, the following statement is not rigorous (L50-L51). The mentioned works (*DDPM* [2] and *DDIM* [3]) weren't designed to impose extra conditions like text (an attention-based denoiser is indeed proposed in *DDPM*); rather, *LDM* [4] is what paved the way for capturing these text-driven image generation. This kind of statement may mislead the readers.
> *Attention-based approaches Ho et al. (2020); Song et al. (2020) have been proposed to better capture relationships between words and visual features,*

4. **Presentation Issues**: The citation style isn't consistent—using `\citep` instead of `\cite`. Additionally, there are quite a few grammatical errors and informal math expressions, making parts of the paper hard to follow.

5. **Inconsistent Terminology**: There’s a discrepancy with how LLM is referenced—in the abstract, it’s “*large language **module***,” but later, it’s called a “*large language **model***.” Consistency here would help avoid confusion and “*large language model*” is more formal.


**Reference**

[1] *Tuo, et al. "Anytext: Multilingual visual text generation and editing." arXiv (2023).*

[2] *Ho, et al. "Denoising diffusion probabilistic models." NeurIPS (2020)*.

[3] *Song, et al. "Denoising Diffusion Implicit Models." ICLR (2021)*.

[4] *Rombach, et al. "High-resolution image synthesis with latent diffusion models." CVPR (2022)*

**Questions:**

1. What exactly is "*Stable Diffusion v1.6*"? From what I know, *Stable Diffusion v1.5* [5] is the most widely used in the 1.x series. If v1.6 exists, could you provide its source or any documentation?

2. In Figure 3, the **second row's last column** shows results under the prompt "*A street sign with 'Main Street' written on it*." The output looks suspiciously like an image from the Hugging Face Diffusers documentation [6]. Can you clarify the validity of this comparison?

3. What about the time efficiency of the proposed method? Since this framework involves multiple stages (like calls to OpenAI API and online search engines), I assume the time cost could be higher. Could you provide a quantitative comparison with other methods?

4. In Table 1, you present quantitative results comparing layout-based methods and LLM-based methods, but the qualitative comparison doesn’t include these approaches. Why are these missing from the visual results?

5. Also in Table 1, there’s an odd repetition of numbers. For example, why do the results for *VPGen*, *Ranni*, and *LMD* seem identical? This repetition needs clarification.


**Reference**

[5] *https://huggingface.co/stable-diffusion-v1-5/stable-diffusion-v1-5*.

[6] *https://huggingface.co/docs/diffusers/en/using-diffusers/inpaint*.

---

### Official Review · Reviewer_Aytd · 2024-10-29

**Soundness:** 3
**Presentation:** 3
**Contribution:** 2
**Rating:** 5
**Confidence:** 4

**Summary:**

This work introduces the Prompt-Consistency Image Generation(namely PCIG), a effective approach that significantly enhances the alignment of generated images with their corresponding descriptions. By leveraging a state-of-the-art large language module to make a comprehensive prompt analysis, this work first generate bounding box for each identified objects. Then it further integrates a state-of-the-art controllable diffusion model with a visual text generation module to generate an image guided by bounding box, which demonstrates the method could handle various type of object category based on the integration of text generation module and search engine. Both qualitative and quantitative results demonstrate the superior performance.

**Strengths:**

The strength of this work lies mainly in the fact that the tasks and questions raised are very important and meaningful. Specifically,
- This paper introduces PCIG, a novel framework that integrates LLMs, knowledge graphs, and controllable diffusion models to generate prompt-consistent images.
- This work proposes a comprehensive approach to address three key aspects of consistency: general objects, text within the image, and objects referring to proper nouns.
- Extensive experiments on a multimodal hallucination benchmark demonstrates the superiority of PCIG over existing T2I models in terms of consistency and accuracy.
- Providing insights into the effectiveness of integrating LLMs and knowledge graphs for prompt understanding and object localization in the image generation process.

**Weaknesses:**

- **The innovativeness and practicality of the solution is somewhat limited.** This work raises a very important issue in the field of image generation - ``*generative hallucination*'', and classifies it into 4 categories and defines it well. However, the solution seems to be to introduce a lot of external components such as large models, diffusion models, knowledge graphs, object extraction models, relation extraction models, etc. to build a pipeline rather than an end-to-end solution. This leads to an important question: if one of the links, such as object extraction or relationship extraction, goes wrong, does it directly lead to model errors? Therefore, I think such a model is not innovative enough and is difficult to use in practice. Its reliability depends on the stability of each component and the adaptability of text prompts. For example, some complex scenarios (omitted subjects, professional terms, abbreviations, etc. are difficult to parse in a long textual prompt)

- **The details of the method are not clearly described.** In sections 3.2 to 3.5 of the method, the author focuses on "what to do" rather than "how to do it", so it is difficult to understand the details of the method, such as: Do the object extraction and relation extraction modules rely solely on a large language model?

- **Whether to support overlapping of hallucination categories.** I think generative hallucination is not a deterministic three-classification problem. It is very likely to have overlaps. For example, in the second example in Figure 1, the visual text presented is likely to need to ensure the consistency of object position, the correctness of scene text recognition, and the alignment with world knowledge (vocabulary or professional terms) at the same time. How would the model cope with it in this case?

- **Questions about the results in Table 1.** How is the TFH indicator calculated and what is its relationship with TH and FH? Why are the TH and FH metrics extremely low in other methods but very high in this work? Is this a fair comparison in the experimental setting and metric calculation? Are the TFHs of other methods all 0? Can the experiment be repeated and what about testing for statistical significance?

**Questions:**

See weaknesses.

---

### Official Review · Reviewer_2YyQ · 2024-11-03

**Soundness:** 2
**Presentation:** 2
**Contribution:** 1
**Rating:** 3
**Confidence:** 5

**Summary:**

This paper aims to address an important problem in current text-to-image generation models: the generated visual contents may not exactly follow the input text description. To solve this problem, this paper proposes a refinement method that first extracts object attribute description and object relation. Then a relation graph is built as input to a controlnet to improve consistency of the generated images with the text description.

**Strengths:**

- The targeted problem is important
- The solution is reasonable and works well in the benchmark evaluation

**Weaknesses:**

- The proposed method does not solve the problem in a fundamental manner (like improve consistency of the image generation model itself). The proposed method involves multiple components. Although the overall pipeline seems reasonable, the involved components may introduce other accumulative errors which may hurt the quality of the generated images.

- The proposed method will introduce extra computation cost for image generation model.

**Questions:**

My main concern is that this approach feels more like a temporary fix for the current T2I models rather than a solution to their fundamental limitations.

How much extra computation cost will be introduced?

How to address the errors introduced by the new components (like the built graph)?

---

### Official Review · Reviewer_qWLq · 2024-11-03

**Soundness:** 2
**Presentation:** 2
**Contribution:** 2
**Rating:** 5
**Confidence:** 5

**Summary:**

The paper proposes PCIG, a diffusion-based framework to significantly enhance the alignment of generated images with their
corresponding descriptions, addressing the inconsistency between visual output and textual input. The framework leverages an LLM to extract key information components from the prompt. This information is then integrates with a state-of-the-art controllable image generation model with a visual text generation module to generate an image that is consistent with the original prompt, guided by the predicted object locations. Extensive experiments have been performed on multimodal hallucination benchmarks.

**Strengths:**

- The paper is easy to follow.
- The proposed can provide further insights into hallucinations problems in generative models.

**Weaknesses:**

- Firstly it seems the paper has been written hastily as I don't see much coherence in introduction section.
- The related work is not comprehensive. The authors have missed many existing related works. For instance, in the "LLM assisted generation" section, authors do not discuss [1] and [2] which are very relevant to the paper.
- The method used by the authors has a very limited novelty as the object extraction, attribute extraction, bbox generation from LLM has already been explored in [1] and [2]. Both these works are solving advanced problems and use this approach just as a small part of their method.
- Again the comparisons with many existing layout based methods is missing such as [1] and [2],
- The comparisons with baselines seem unfair. The proposed method uses GPT-4o Turbo while baselines either use existing open-source LLMs or other version of GPT (such as LayoutGPT) or do not use LLM at all. I do not think the scores in Table 1 are valid with these diverse settings.
- The authors should also compare their works with Prompt refinement papers such as  [3] and [4].
- Overall while the paper presents a good direction, there are major issues with respect to novely, missing comparisons and fairness of comparisons.



Reference

[1] Gani, Hanan, et al. "Llm blueprint: Enabling text-to-image generation with complex and detailed prompts." ICLR (2024).

[2] Lian, Long, et al. "Llm-grounded diffusion: Enhancing prompt understanding of text-to-image diffusion models with large language models." TMLR 2024.

[3] Hao, Yaru, et al. "Optimizing prompts for text-to-image generation." NeurIPS 2024.

[4] Zhan, J., Ai et al. Prompt Refinement with Image Pivot for Text-to-Image Generation. ACL 2024.

**Questions:**

- Can the proposed method generalize to long and complex prompts?

---

### Note · Authors · 2024-11-14

I have read and agree with the venue's withdrawal policy on behalf of myself and my co-authors.